# A Novel Knowledge Base Question Answering Method Based on Graph Convolutional Network and Optimized Search Space

**Xia Hou \*, Jintao Luo, Junzhe Li, Liangguo Wang and Hongbo Yang**

Computer School, Beijing Information Science & Technology University, Beijing 100101, China
* Correspondence: houxia@bistu.edu.cn

**Abstract:** Knowledge base question answering (KBQA) aims to provide answers to natural language questions from information in the knowledge base. Although many methods perform well when dealing with simple questions, there are still two challenges for complex questions: huge search space and information missing from the query graphs' structure. To solve these problems, we propose a novel KBQA method based on a graph convolutional network and optimized search space. When generating the query graph, we rank the query graphs by both their semantic and structural similarities with the question. Then, we just use the top k for the next step. In this process, we specifically extract the structure information of the query graphs by a graph convolutional network while extracting semantic information by a pre-trained model. Thus, we can enhance the method's ability to understand complex questions. We also introduce a constraint function to optimize the search space. Furthermore, we use the beam search algorithm to reduce the search space further. Experiments on the WebQuestionsSP dataset demonstrate that our method outperforms some baseline methods, showing that the structural information of the query graph has a significant impact on the KBQA task.

**Keywords:** knowledge base question answering; query graph; question answering; knowledge base

## 1. Introduction

A knowledge graph is a heterogeneous multi-digraph, which means it is directed, and multiple edges can exist between two nodes. An agent generates knowledge by relating elements of a graph to real-world objects and actions. A knowledge graph (KG), also known as a knowledge base (KB), is a structured representation of facts that describes a collection of interlinked descriptions of entities, relationships, and semantic descriptions of entities [1]. Knowledge bases store a large amount of factual knowledge from the real world. Many large KBs, such as DBPedia [2], Freebase, YAGO [3] and NELL [4], have been built to serve downstream tasks. Knowledge base question answering (KBQA), which aims to answer natural language questions by knowledge bases, has received a lot of attention as an important research direction [5–8]. Figure 1 shows the process of finding the answer to a question by the knowledge in KB.

Semantic parsing-based methods (SP-based methods) are one of the mainstream approaches for KBQA [9,10]. The SP-based methods first convert natural language questions into symbolic logical forms; after that, the answers are obtained by executing them in a knowledge base [11]. Such methods can visualize the reasoning process, which makes the results have high interpretability. However, they rely heavily on the design of logical forms and parsing algorithms.

Some works combine graphical structures with SP-based methods to solve the problem [12,13]. These methods transform question answering into a query graph generation process and show powerful expressive power in the complex KBQA task. However, such approaches still face two problems. (1) The number of query graphs grows exponentially with the growth of the knowledge base size and the emergence of complex questions [14].

(2) Most works only consider the semantic information of the query graphs, while they ignore the natural graph structure features. However, the latter information is also useful for selecting the correct query graphs [15]. Therefore, how to reduce the number of candidate query graphs and how to precisely select the correct query graphs are still the key challenges of the current KBQA work.

Question: In which stadium did Player A's team win the 1998 World Championship?

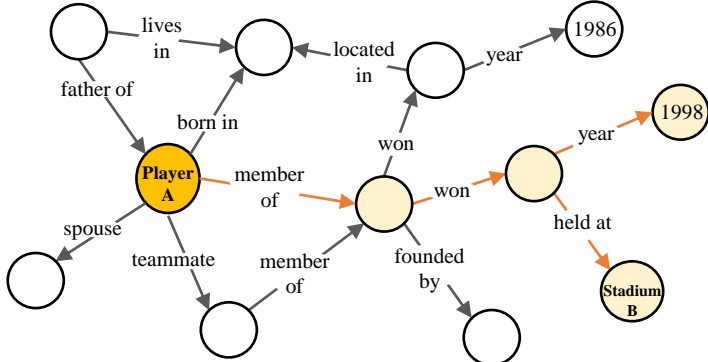

**Figure 1.** An example of a KBQA task. For the question "In which stadium did Player A's team win the 1998 World Championship?", the orange circle and the orange line represent the inference process from Player A (the topic entity) to Stadium B (the answer).

In this paper, we focus on how to address the two challenges. For challenge 1, we observe that usually, the correct answer to a complex question cannot be found just once in the large search space. Therefore, we can use staged queries to decompose complex questions into multiple simple questions. In addition, a complex question has more than one constraint, which can be used to further reduce the search overhead. We note that some approaches use the graph structure information to improve the effect in some other Natural Language Processing (NLP) tasks but not KBQA [16,17]. In fact, the structure of the query graph is also useful for KBQA. Therefore, for challenge 2, we extract the structure information of the query graphs to enhance the ability of our method to select the correct answer in KBQA.

Based on the motivation above, we propose a novel KBQA method based on a graph convolutional network by optimized search space. We transform the process of answering complex questions into a hierarchical process of generating query graphs. We extract the constraint function from the complex question and use it to reduce the number of candidate query graphs. After that, we design a novel ranker that scores the candidate query graphs using two components: semantic similarity matching and graph structural similarity. Finally, it uses the beam search algorithm to select the Top K highest-rated query graphs from the candidate query graphs. Due to the addition of the graph structure similarity matching module, our method can select query graphs more accurately. Our main contributions are as follows:

1. To reduce the huge search space for KBQA, we use a constraint function as well as the beam search algorithm to limit the number of candidate query graphs and reduce the computational overhead.
2. To update the correctness of query graphs, we add structural information to the semantic information of the query graphs and score the query graphs from multiple perspectives, which enhances the model's ability to understand complex questions.
3. Experimental results on the publicly available KBQA dataset WebQuestionsSP show that our method achieves good experimental results compared to the baseline methods.

## 2. Related Work

### 2.1. Semantic Parsing-Based Methods for KBQA

Semantic parsing-based methods are the most dominant class of KBQA methods, which aim to parse natural language discourse into logical forms [18,19]. Specifically, this category of methods first encodes the question through semantic and syntactic analysis. Afterward, the encoded questions are converted into logical forms of statements (e.g., SPARQL Protocol and RDF Query Language (SPARQL) and Structured Query Language (SQL)) by using a logical parsing module. Finally, the obtained logical form statements are executed on the knowledge base to query the answers [20,21].

The earlier methods [22,23] can handle simple questions well. However, in the subsequent large-scale knowledge bases, these traditional methods are no longer applicable in the face of complex questions with complex semantic syntax involving multiple entities.

### 2.2. Query Graph-Based Methods for KBQA

The concept of query graph was first proposed by Yih et al., 2015 [12], which is a new idea to simplify the traditional semantic parsing-based methods [13,14]. The query graph-based method introduces the semantic information formed by entities and relations in the knowledge base during the parsing of a question. It transforms the semantic understanding process of a question into a query graph generation process, which shows the semantic matching process more intuitively and thus has very good interpretability.

However, the query graph generation process usually relies on predefined manual rules, which are not well suited for a large number of complex questions in a large-scale knowledge base. To alleviate this, Ding et al., 2019 [24] used the substructure of frequently occurring queries to assist query graph generation. Abujabal et al., 2017 [25] automatically generated templates based on question–answer pairs to reduce manual operations. Hu et al., 2018 [26] applied aggregation operations and coreference resolution techniques to accommodate complex questions.

In addition, earlier methods only consider the degree of predicate matching in the natural language question and the query graph. They use the core query path in the query graph to measure the similarity to the question [12,27]. These methods omit much useful information and lead to less accurate filtering of the query graph. Based on this, Lan et al., 2020 [28] more comprehensively utilized the information from nodes, relations, and constraints in the query graph generation process. They transformed the query graph into a serialized form containing nodes, relations, and constraints before performing the semantic similarity measure, which enhances the matching ability of their method to the correct query graph. However, the serialization process causes two nodes that are originally adjacent to each other to be split in the sequence, distorting part of the semantic information and destroying the graph structure information that the query graph naturally has.

## 3. Method

### 3.1. Overview of the Method

**Task Description:** A KB collects knowledge data in the form of triples $K = \{h, r, t\}$, where $r \in R$ (the set of relations) and $h, t \in E$ (the set of entities). For a given natural language question $q$, the KBQA task is to find the answer $a$, where $a \in E$.

**Method overview:** We propose a novel KBQA method based on a graph convolutional network by an optimized search space. We formalize the KBQA task as maximizing the probability distribution $p(a|K, q)$. Instead of reasoning directly about $K$, we retrieve a query graph $g \in K$ and infer $a$ on $g$. Since $g$ is unknown, we treat it as a latent variable and rewrite $p(a|K, q)$ as:

$$p(a|K, q) = \sum_{g} p(a|q, g) p(g|q) \tag{1}$$

To obtain the query graph $g$, our method starts from the topic entity in question $q$ and generates the query graph hierarchically using the *extend* or *constrain* operations, which are described in Section 3.2.

We assume that the correct query graph has a high degree of similarity to the question $q$. We can use this to select the correct path from the generated candidate query graphs. To measure this similarity, we design a *Ranker* (described in Section 3.3) that selects the candidate query graphs based on semantic matching and structural similarity of the graphs.

Specifically, we use the pre-trained language model RoBERTa to measure the semantic similarity between the question $q$ and the candidate query graphs. At the same time, we use a graph convolutional network to encode the semantic and structural information of the candidate query graphs together, after which we can measure the similarity of these candidate query graphs. Finally, we combine a constrain function and a beam search algorithm to select the query graphs with high similarity for the next step. The beam search algorithm improves the greedy search algorithm by selecting *beam* − *size* candidates from the set of candidates generated by each search as the starting point for subsequent searches. Therefore, we can select the *beam* − *size* query graphs with high similarity scores from all candidate query graphs, which largely reduces the number of query graphs to optimize the search space.

We repeat the above generation-ranking operation until we find the correct answer or reach the maximum hop count limit. An example in Figure 2 shows the process of our method to find the correct answer to a question.

**Figure 2.** An example of the query process. Starting from a topic entity in the question, the *extend* and *constrain* operations are applied to generate the query graphs and eventually find the answer. The orange circles represent the constraint function to reduce the search space. *Ranker* is used to select the path with a higher score after ranking the candidate paths, such as the path made up of the orange arrow and the lambda variable $X$.

### 3.2. Query Graph Generation

This module uses two actions: *extend* and *constrain* to generate query graphs.



The *extend* action extends the core relational path by adding relations (selected by *Ranker*) to the query graph. Specifically, we connect the relation $r$ chosen by *Ranker* to the lambda variable $X$ (or the topic entity $e_t$). After the connection, the original lambda variable $X$ becomes an intermediate variable $y$ (the topic entity $e_t$ remains unchanged), while the other end of $r$ becomes the new lambda variable $X$.

Referring to Luo et al., 2018 [29], we generate a constraint function by matching the keywords (e.g., first, last, biggest, etc.) in the question. The *constrain* action attaches the detected constraint function to the lambda variable $X$ or an intermediate variable connected to $X$. In the example in Figure 2, when our method detects the keyword 'first', it generates a constraint function *argmin*, which limits the search to the nodes around which it is connected. Such a constraint helps the model limit the search to a certain range, which reduces the search space.

This module starts with the topic entity (topic entity linking results are from the paper [28]) and uses the *extend* action or *constrain* action to generate the query graph step by step. Some previous methods [12,27] place the process of adding constraints after the core path is fully generated. However, such methods are too simple and have a limited reduction in the number of candidate query graphs. Therefore, our method performs the *constrain* action before the *extend* action, which reduces the number of candidate query graphs.

### 3.3. Query Graph Ranker

For the method that uses enumeration to search [12], the number of candidate query graphs approaches $k^n$, where $k$ is the core path length and $n$ is the average number of single-hop candidate paths. For complex questions, $k^n$ varies from thousands to millions. Such an order of magnitude cannot be handled with current methods.

Therefore, to prevent the number of candidate query graphs from growing exponentially with the number of query steps, we use a beam search algorithm to limit the number of query graphs obtained at each step. Further, in order to select query graphs associated with the correct answers, we design a scoring function to rank the query graphs by both semantic and graph structure perspectives of the query graphs and some simple features. Figure 3 is an example of the query graph *Ranker*.

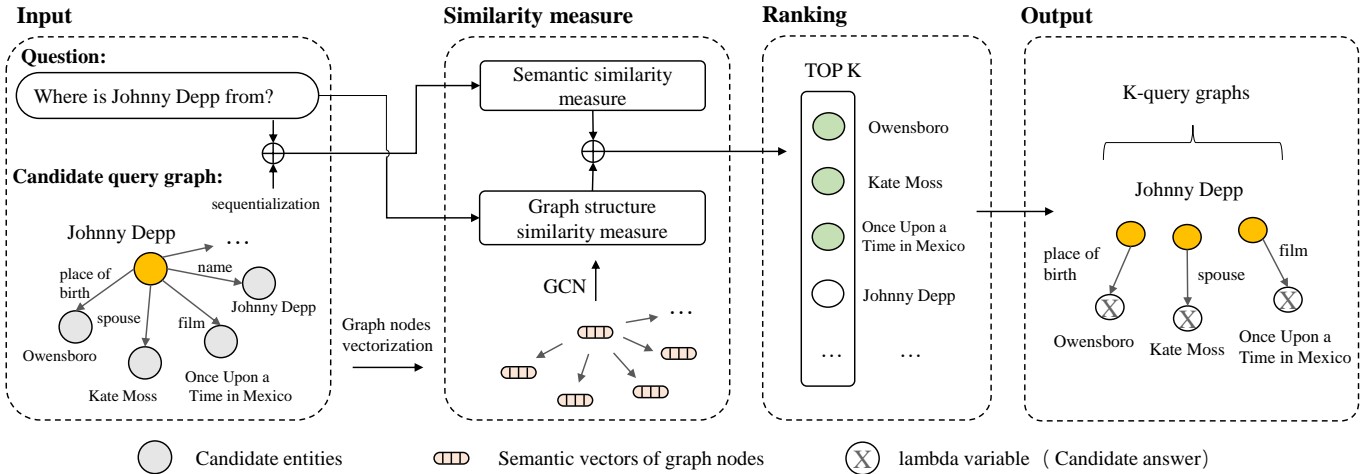

**Figure 3.** The structure of the query graph *Ranker*.

### 3.3.1. Semantic Similarity Measure

This module aims to measure the semantic similarity of the natural language question $q$ and the query graph $g$. This module starts from the topic entity in the question and transforms the query graph into a sequence form $g'$ containing entities and relations according to the query graph generation process.

Specifically, we compose the question $q$ and the query graph sequence $g'$ into a statement pair as the input to RoBERTa (robustly optimized BERT approach) [30]. Then, their semantic similarity $score(q, g')$ is obtained. The formulas are as follows:

$$H_{qg'} = RoBERTaCLS([q; g']) \tag{2}$$

$$score(q, g') = LINEAR(H_{qg'}) \tag{3}$$

where RoBERTaCLS denotes the (CLS) representation of the concatenated input (Figure 4), and $LINEAR$ is a projection layer reducing the representation to a scalar similarity score.

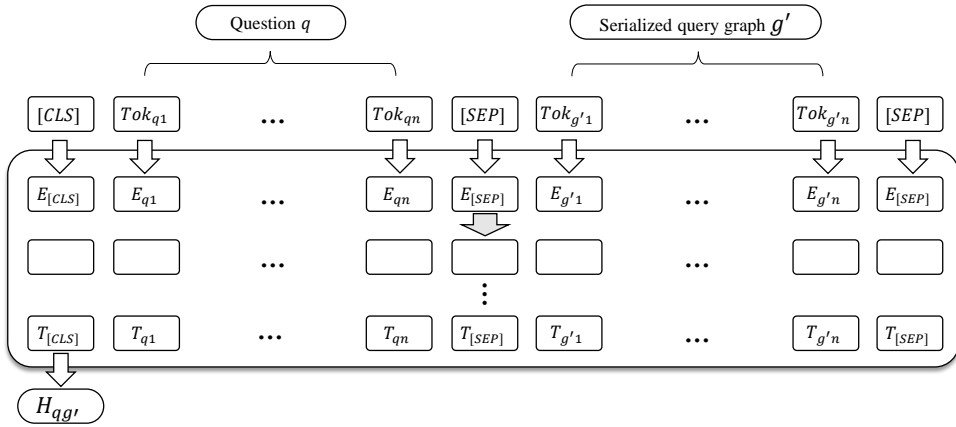

**Figure 4.** The input and output of RoBERTa for measuring the semantic similarity.

3.3.2. Graph Structure Similarity Measure

The semantic similarity metric module lacks the structural information of the query graph. Furthermore, the sequence transformation process leads to the segmentation of adjacent nodes in the query graph. Therefore, in addition to the semantic information mentioned in Section 3.3.1), this module also parses the query graph from the view of its structure.

First, the module vectorizes a node and its type as $Ne$ (using the Global Vectors for Word Representation (GloVe)). After that, $Ne$ is fed into the Bi-directional Long Short-Term Memory (Bi-LSTM), and the hidden state $h_e$ of the last time step of the Bi-LSTM is selected as the final encoding of the node, i.e.,

$$h_e = Bi - LSTM(Ne) \tag{4}$$

At this point, the initial description of each node in the query graph is obtained, but each node in the current graph contains only its own information and lacks the description of its neighboring nodes. Therefore, this module uses the Graph Convolutional Network (GCN) to represent the query graph $g$. The GCN hierarchically aggregates nodes and their neighbor representations. After several aggregations, the nodes contain more information about their neighborhoods. Then, $h_g$, the final representation of the graph $g$, is obtained by averaging over all nodes' representations. The formulas are as follows:

$$h_i^{(l+1)} = ReLU\left(\sum_{j \in N(i)} \frac{h_j^{(l)}}{D_{ji}} W^{(l)} + b^{(l)}\right) \tag{5}$$

$$D_{ji} = \sqrt{|N(j)|}\sqrt{|N(i)|} \tag{6}$$

$$h_g^{(l)} = \frac{1}{|V|} \sum_{i \in V} h_i^{(l)} \tag{7}$$

where $N(i)$ is the set of neighbor nodes of node $h_i$; $h_j^{(l)}$ is the representation of node $h_j$ in the $l$-th iteration; $W^{(l)}$ is the parameter matrix of each layer of linear transformation; $b^{(l)}$ is the bias value of each aggregation; and $V$ denotes the set of nodes in graph $g$.

Finally, the graph structure similarity $score(q, g)$ is measured by using the cosine similarity:

$$score(q, g) = cos(h_q, h_g) \tag{8}$$

where $h_q$ is the vector representation of the question $q$ (obtained from RoBERTa).

### 3.3.3. Candidate Query Graph Selection

We design a scoring function that uses the previously obtained semantic similarity and structure similarity as well as some simple features as evaluation criteria to rank the candidate query graphs, the formulas are as follows:

$$Features = score(q, s) \oplus score(q, g) \oplus F_{answer} \oplus F_{topic} \oplus F_{cons} \tag{9}$$

$$SCORE = sigmoid(W[Features] + b) \tag{10}$$

where $F_{answer}$ is the number of candidate answers; $F_{topic}$ is the topic entity score; $F_{cons}$ is the number of constraints; and $W$ and $b$ parameters are to be learned during model training.

Finally, we use the beam search algorithm to select the top K candidate query graphs for the next iteration.

## 4. Experimentals

### 4.1. Datasets

We conduct experiments using the WebQuestionsSP (WebQSP [31]) dataset to evaluate the effectiveness of our method. WebQuestionsSP is a widely used publicly available dataset containing 4737 questions based on Freebase KB. Following Sun et al., 2018 [32], we partitioned the dataset into the training/validation/testing sets with the number of 2848/250/1639 questions.

### 4.2. Methods for Comparison

We have selected several methods in related fields within the last few years as baseline methods. First, we compare the method proposed by Lan et al., 2019 [33], which considers the complexity of multi-hop relational paths but does not use set searches or constraints to reduce the search space. After that, we compare the method of Chen et al., 2019 [34], who transforms the extraction of multi-hop relationships into multiple single-pick extractions, thus reducing the search space. We also compare the method that uses additional information: Han et al., 2020 [35] take textual information as hyper-edges and update entity states using GCN. Next, we compare the method of Yan et al., 2021 [36] that uses auxiliary tasks to enhance the pre-trained model. Then, we compare the method of Qin et al., 2021 [37], who use the relational graph to reduce the search space of the query graph. Finally, we compared some of the latest methods [7,8,14,38]. Among them, Zhang et al., 2022 [7] composed subgraphs from multiple entities. Chen et al., 2022 [14] used abstract query graphs to enhance query graph accuracy. Ye et al., 2022 [8] and Hu et al., 2022 [38] used generative methods to find answers.

### 4.3. Results

The results of our method compared with the baseline methods on WebQuestionsSP are shown in Table 1.

The method of Qin et al., 2021 [37] reduces the number of candidate query graphs but does not extract the graph structure information of query graphs. Although Han et al., 2020 [35] use GCN to extract graph structure information, they ignore the matching of semantic information. Yan et al. 2021 [36] reformulate the retrieval-based KBQA task to

make it a question-context matching form and propose three auxiliary tasks for relation learning, namely relation extraction, relation matching, and relation reasoning, which gives the best results (Hit@1-score) among all baseline methods. Due to the clear supervised signal, these supervised models show excellent performance. In particular, the method of Ye et al., 2022 [8] achieved a surprising F1-score of 76.6.

In contrast, our method not only extracts semantic information by using a pre-trained model but also uses GCN to extract graph structure information. Furthermore, we also combine the beam search algorithm and constraint function to enhance the performance of our method. Thus, our method achieves competitive performance on the WebQSP dataset compared to other baseline methods.

**Table 1.** Experimental results for comparison with baselines.

| Method | F1 | Hits@1 |
| --- | --- | --- |
| Lan el al. (2019) [33] | 67.9 | 68.2 |
| Chen et al. (2019) [34] | 68.5 | - |
| Han et al. (2020) [35] | 60.6 | 68.4 |
| Yan et al. (2021) [36] | 64.5 | **72.9** |
| Qin et al. (2021) [37] | 66 | - |
| Zhang et al. (2022) [7] | 64.1 | 69.5 |
| Chen et al. (2022) * [14] | 70.3 | 70.6 |
| Ye et al. (2022) * [8] | 75.6 | - |
| Hu et al. (2022) * [38] | **76.6** | - |
| our method | 68.9 | 68.5 |

* denotes supervised methods that use gold SPARQL (or ground truth logical form) as a supervised signal. Our method uses only question–answer pairs, which is a weakly supervised method. The bolded scores represent the highest scores.

### 4.4. Ablation Study

In order to verify the validity of each component in the model, we performed an ablation study. Table 2 shows the experimental results.

**Table 2.** Experimental results of the ablation study.

| Mothod | F1 | ΔF1 |
| --- | --- | --- |
| our method | 68.9 | 0.0 |
| w/o RoBERTa | 62.9 | −6.0 |
| w/o GCN | 66.7 | −2.2 |
| w/o Other features | 68.3 | −0.6 |

**Variant 1** (w/o RoBERTa): We use Gate Recurrent Unit (GRU) to replace RoBERTa in the model. The performance of the model decreased by 6.0% due to the prevalence of missing links in the knowledge base. For example, 71% of the person entities in Freebase are missing birthplace information [39]. This leads to the fact that two logically related nodes are not linked in the knowledge base, which reduces the likelihood of finding the correct answer. However, the pre-trained model contains knowledge of many open domains and can make predictions about the missing links in the KB.

**Variant 2** (w/o GCN): We remove part of the graph structure similarity measure. The performance of the model decreases by 2.2%, which confirms that for query graph-based KBQA methods, extracting the graph structure of the query graph is important. The query graph cannot be filtered well by semantic information matching alone.

**Variant 3** (w/o Other features): We remove the simple features of the candidate query graph selection module. This variant has the lowest performance degradation of 0.6%. This proves that these simple features are much less capable of filtering query graphs than semantic matching as well as graph structure matching.

Furthermore, in order to evaluate the impact of the model components more extensively, we continued our experiments based on **Variant 1**. The results are shown in Table 3.

**Table 3.** Ablation study for **Variant 1**.

| Method | F1 | ΔF1 |
| :---: | :---: | :---: |
| **Variant 1** | 62.9 | 0.0 |
| w/o GCN | 60.5 | −2.4 |
| w/o Other features | 62.4 | −0.5 |

**Variant 1-a** (w/o GCN): We removed the graph structure extraction and matching module from variant 1. In this case, the model uses only semantic similarity to select candidate query graphs. The results of the model decreased by 2.4%. This demonstrates that graph structure matching can improve the performance of the query graph-based KBQA model very well.

**Variant 1-b** (w/o Other features): We removed the simple features from variant 1. The model effect is reduced by 0.4%.

The results of these two variants demonstrate that both graph structure and simple features can have some improvement on the KBQA task under different settings.

We also compared the change in the F1-score score during training for each variant (excluding the variant with simple features removed since the difference in their effectiveness was not significant). As can be seen from Figure 5, although the graph structure metric makes the model fluctuate more sharply in the early stages, which makes the model less effective than the variants without considering the graph structure at some point, it also gives the model a higher upper limit.

The ablation study results prove that each module in our model improves the effectiveness of the model. Moreover, the above variants still outperform some of the baseline models, which proves that the effectiveness of our method comes not only from the individual modules but also depends on the overall process design of the model.

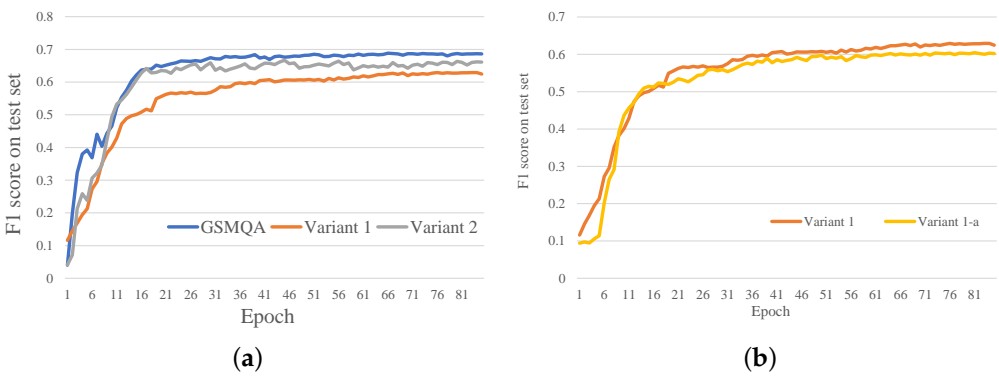

**Figure 5.** Comparison of the F1-score for each variant. (**a**) Comparison of our method and its variants. (**b**) Comparison of **variant 1** and its variant.

## 5. Conclusions

In this paper, we propose a novel KBQA method based on graph convolutional networks and optimized search spaces. By constraining the search process, the model is able to handle complex problems with multiple hops. It solves the problem of graph structure information missing in previous query graph-based KBQA methods, and the results show that the addition of the graph structure matching module improves the model performance by 2.2% (F1-score). Experiments on the WebQSP dataset show that our method has excellent performance.

**Limitations:** In the process of using keywords to detect constraint functions, there may be ambiguity issues. In addition, while using graph structures to improve model performance, our approach leads to an increase in model training time. Further, the large-scale pre-training of the model implies a large resource overhead.

**Future work:** We plan to optimize the model, reduce the resource overhead, and resolve the ambiguity in the constraint function. We also intend to study the effect of different dataset partitions on the experiment.

**Author Contributions:** Conceptualization, X.H. and J.L. (Junzhe Li ); Data curation, J.L. (Jintao Luo), J.L. (Junzhe Li) and H.Y.; Formal analysis, J.L. (Jintao Luo); Investigation, J.L. (Junzhe Li); Methodology, X.H., J.L. (Jintao Luo) and J.L. (Junzhe Li); Project administration, X.H.; Resources, X.H., H.Y. and L.W.; Software, J.L. (JintaoLuo) and J.L. (Junzhe Li); Supervision, X.H., H.Y. and L.W.; Visualization, X.H. and J.L. (Jintao Luo); Writing—original draft, X.H. and J.L (Jintao Luo); Writing—review and editing, X.H., H.Y. and L.W. All authors have read and agreed to the published version of the manuscript.

**Funding:** This research was supported by: Undergraduate Teaching Reform and Innovation Project of Beijing Higher Education, China, Grant Number: 5112210807, and by the project of Excellent teaching management personnel in Beijing universities, Grant Number: 5112210823.

**Data Availability Statement:** The WebQuestionsSP dataset can be accessed via the following link (http://aka.ms/WebQSP).

**Conflicts of Interest:** The authors declare no conflict of interest.

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
