# Peer review of "A Novel Knowledge Base Question Answering Method Based on Graph Convolutional Network and Optimized Search Space"

_electronics, doi:10.3390/electronics11233897_

Round 1

Reviewer 1 Report

please read the attached revision.

Reviewer 2 Report

Thanks for giving me a chance to review this paper. I suggest the following changes.

1.  Add more literature to sections 2. 1 and 2.2

2.  Include limitations and future work

3. Also add research significance and utilization.

Reviewer 3 Report

This study investigates “A Novel Knowledge Base Question Answering Method based on Graph Convolutional Network and Optimized Search Space.”
The topic is interesting and fits well with the scope of the journal. Overall, the manuscript is well written, and some revisions should be made before considering acceptance.

Moreover, I believe the authors should explain their research methods more clearly and deeply.

1. English should be checked. Check grammar and functions (e.g., page 3, line 79).
2. There needs to be more discussion of the research gap. The authors should improve the introduction section by scientifically and clearly emphasizing the research gap and originality.
3. The abstract should be enhanced by including information about research methods, results, and conclusions.
4. It will fine add the published year of the citations (e.g., Yih et al., XXXX ).
5. There should be more scientific explanation in Section 3.1, Overview of the Method.
6. Abbreviations should be mentioned for the first time. Please check and revise the entire manuscript.
7. Section 4.1 “Following Sun et al. [29], we partitioned the dataset into the training/validation/testing set with the number of 2848/250/1639 195 questions.”
i) Are these numbers appropriate for your study? Did you check?
ii) How about the validation outcome?
8. The conclusion should be enhanced with value added.

Round 2

Reviewer 1 Report

well revised.

Reviewer 3 Report

I believe the manuscript is suitable for publication.